# Loss of Cdc42 in Exocrine Acini Decreases Saliva Secretion but Increases Tear Secretion—A Potential Model of Exocrine Gland Senescence

**DOI:** 10.3390/ijms242417220

**Published:** 2023-12-07

**Authors:** Haruna Nagase, Akiko Shitara, Yuta Ohno, Keitaro Satoh, Masanori Kashimata

**Affiliations:** 1Department of Pharmacology, Asahi University School of Dentistry, 1851-1 Hozumi, Mizuho, Gifu 501-0296, Japan; 2Department of Pharmacology, Meikai University School of Dentistry, 1-1 Keyakidai, Sakado, Saitama 350-0283, Japan

**Keywords:** Cdc42, hyposecretion, hypersecretion, saliva, tear, lacrimal gland, salivary gland, AQP5, acinar cell, apoptosis

## Abstract

Cdc42 is a small GTPase essential for the cell cycle, morphogenesis, and cell adhesion, and it is involved in the polarity of epithelial cells. However, the functional roles of Cdc42 in exocrine glands, such as the maintenance of acini and water secretion, are not yet well understood. In this study, we generated acinar-cell-specific *Cdc42* conditional knockout (*Cdc42*^cKO^) mice to assess their maintenance of acinar cells and physiological functions in the salivary glands (SGs) and lacrimal glands (LGs). Our data revealed that the loss of Cdc42 altered the luminal structures to bulging structures and induced acinar cell apoptosis in both the parotid glands (PGs) and LGs of *Cdc42*^cKO^ mice. Interestingly, saliva secretion in response to pilocarpine stimulation was decreased in the *Cdc42*^cKO^ group, whereas tear secretion was increased. Consistent with the water secretion results, protein expression of the water channel AQP5 in acinar cells was also decreased in the PGs but conversely increased in the LGs. Moreover, the changes that increased AQP5 expression in LGs occurred in the acinar cells rather than the duct cells. The present study demonstrates that Cdc42 is involved in the structural and survival maintenance of acinar cells in SGs and LGs. On the other hand, depletion of Cdc42 caused the opposite physiological phenomena between PGs and LGs.

## 1. Introduction

Water secretion is essential for the elimination of substances from the body and is the basis of the secretions of the exocrine glands, such as the salivary, lacrimal, mammary, and pancreatic glands. Among them, the salivary gland (SG) and lacrimal gland (LG) have been compared in various studies, such as tear and saliva contents after irradiation of head and neck cancers and in Sjögren’s syndrome patients and tissue damage [1,2]. That is because they are anatomically, histologically, and physiologically very similar in both humans and mice, are in close facial proximity to each other, and have very similar post-irradiation responses and pathophysiology of Sjögren’s syndrome [3,4]. Common structures of exocrine glands include acini and ducts. Saliva is released from the lumen of the SG acini into the oral cavity through ducts and contributes to oral health by protecting and lubricating the oral mucosa [5,6]. Tears are similarly released from the lumen of the LG acini through a duct into the conjunctival sac, where they protect the ocular surface [7]. Because these secretions drain into the lumen through the apical membrane of the acinar cells, the directionality of the basal and apical sides of the acinar cells—or epithelial cell polarity—is important for this water secretion [8]. Although both SGs and LGs are similarly responsible for water secretion, there might be some differences in their secretory functions since it is not well known whether these cell polar properties are exactly the same in both glands. What is the same and what is different in water secretion mechanisms between different exocrine glands remains unclear.

Cdc42, a member of the Rho family of small GTPases, is known to play diverse roles in various cell types, including the cell cycle, adhesion, morphology, and polarity, and is associated with cell growth, migration, endocytosis, and exocytosis [9]. However, most studies were conducted using cell cultures and small organisms because systemic KO of Cdc42 results in embryonic lethality [10]. Therefore, we have a poor understanding of whether Cdc42 has similar functions in animal tissues and what effects it has on physiological functions in live mammals.

We previously analyzed the role of Cdc42 in the submandibular glands (SMGs) composed of epithelial cells using mouse models. Our study demonstrated the importance of Cdc42 in maintaining the apical membrane of SMGs through *Cdc42* knockout induced by retrograde injection of adenoviral vectors [11]. However, due to the experimental limitations of virus-based knockout (KO), KO was induced in only some acinar cells. The technique is suitable for studying the properties of individual cells, but not for tissue-wide functional analysis (e.g., secretion levels). Therefore, it remains unclear whether Cdc42 in acinar cells of mature exocrine glands regulates water secretions such as saliva.

In this study, we used tamoxifen-induced *Cdc42* conditional knockout (*Cdc42*^cKO^) mice to evaluate the physiological functions of these two different exocrine glands, LGs and SGs. Tamoxifen-induced Cre/loxP is widely used for gene regulation to enable gene control both spatially and temporally. It is an essential tool when early deletion or overexpression can cause developmental defects or embryonic lethality. Mist1 is a transcription factor that is expressed in the acinar cells of exocrine glands [12,13,14]. *Mist1*^CreERT2^ mice have a creER^T2^ fusion gene replacing the entire coding region of the basic helix-loop-helix family member a15 gene. Restricted to the cytoplasm, Cre-ER^T2^ can only gain access to the nuclear compartment after exposure to tamoxifen. When these mice are bred with mice containing loxP-flanked sequence, tamoxifen-inducible, Cre-mediated recombination will result in deletion of the floxed sequences in the Cre-expressing cells of the offspring. To breed *Mist1*^CreERT2^ mice with *Cdc42* floxed mice which have a loxP sequence on exon2 will enable KO of *Cdc42* specifically in the acinar cells of the exocrine glands. Using these mice, we are addressing the research question of whether Cdc42 regulates acinar cell maintenance and water secretion in SGs and LGs. First, we examined *Cdc42* mRNA and the luminal structure to evaluate the level of Cdc42 KO. Second, we measured gland weights and apoptosis to analyze how Cdc42 KO affects the overall tissue structure. Finally, we measured the volumes of saliva and tear secretion and the expression of AQP5, a water channel. Our results demonstrate a novel molecular function for Cdc42 and suggest that the process of water secretion via regulation of AQP5 expression differs in each exocrine gland.

## 2. Results

### 2.1. Acinar Cell-Specific Cdc42 Conditional Knockout Mice Had Altered Luminal Structures in SGs and LGs

To assess the contribution of acinar cell Cdc42 to the secretion function of mature exocrine glands, we generated tamoxifen-induced, acinar cell-specific, *Cdc42* conditional knockout mice. *Mist1*^CreERT2/CreERT2^/*Cdc42*^fl/fl^/mTmG^+/+^ mice were crossed with *Cdc42*^fl/fl^/mTmG^+/+^ mice to generate *Mist1*^CreERT2/+^/*Cdc42*^fl/fl^/mTmG^+/+^ mice with *Mist1* heterozygosity (Figure 1a). To induce deletion of *Cdc42* which contains loxP sequence on exon2 in adult mice via tamoxifen-inducible Cre/loxP recombination, we administered tamoxifen to these mice and produced acinar cell-specific *Cdc42*^cKO^ mice. As predicted, quantitative RT-PCR analysis showed reduced expression of *Cdc42* genes in the glands of tamoxifen-treated mice (Figure 1b). The body weights of mice were measured before and after corn oil or tamoxifen administration, but no differences were observed between the two groups (Figure 1c).

In our previous study, we showed that infection-positive acinar cells lacking Cdc42 in an adenovirus vector-based system have disrupted apical membranes in their SGs [11]. In this study, we used *Mist1*-Cre mice to examine the effects of Cdc42 loss on gland luminal structures in the majority of acinar cells. To examine whether Cdc42 maintains the apical membrane, sections of mature PGs, SMGs, sublingual glands (SLGs), and LGs were stained with F-actin and merged with mTmG images tracking Cdc42 deficiency. Mist1 is not expressed in ductal cells, but in acinar cells in each of the three major SGs. As expected, the plasma membranes of the acini changed from red to green fluorescence. Mist1 is also known to be expressed in the acinar cells of LGs [13,14]. The luminal structure of an LG, another exocrine gland from the same individual in which we examined 3 major SGs, was also found to be expanded (Figure 1d), indicating that the localization of *Cdc42*^cKO^ acini were confirmed. These data demonstrate that tamoxifen-induced acinar cell-specific *Cdc42*^cKO^ in mice is feasible. 

### 2.2. Cdc42^cKO^ Reduces the Number of Acinar Cells in Exocrine Glands by Inducing Apoptotic Cell Death

Next, the tissue weights of SMGs, PGs, SLGs, and LGs were measured to evaluate the effects on overall tissue structure. The size of all *Cdc42*^cKO^ glands was smaller than in the age-matched control groups. Despite no change in body weight, the weights of PGs, SMGs, and LGs were significantly reduced in *Cdc42*^cKO^ mice compared with the control groups (Figure 2a). This led us to consider the possibility that the acinar cells were undergoing cell death, so we used TUNEL staining to estimate if acinar cells were dying by apoptosis. For subsequent experiments, we decided to compare PGs with the LGs as a representative of three major SGs, since they have comparable Cdc42 KO levels based on our results of *Cdc42* qPCR (Figure 1b), and the saliva secreted by PGs is serous and the tears secreted by LGs are also serous [1,15]. TUNEL-positive nuclei increased in acinar cells in the *Cdc42*^cKO^ groups of both exocrine glands. This suggests that *Cdc42*^cKO^ induces cell death by apoptosis and thereby decreases the number of acinar cells in exocrine glands (Figure 2b). Taken together, these data suggest that Cdc42 in mature acini is important in the maintenance of acinar cells in both SGs and LGs.

### 2.3. Cdc42 Depletion Decreased Pilocarpine-Stimulated Saliva Secretion, but Increased Tear Secretion

To analyze the functions of Cdc42 in regard to gland secretions, we examined saliva and tear secretion in response to stimulation with pilocarpine. The baselines of saliva and tear secretions before pilocarpine stimulation were similar in the *Cdc42*^cKO^ and control groups. The volume of saliva secretion stimulated by pilocarpine significantly decreased in *Cdc42*^cKO^ (Figure 3a,b). However, the volume of tears secreted significantly increased in *Cdc42*^cKO^ (Figure 3c,d). These findings reveal that Cdc42 depletion reduced pilocarpine-stimulated saliva secretion, while increasing tear secretion.

### 2.4. AQP5 Protein in Acinar Cells Is Downregulated in PGs but Upregulated in LGs

To elucidate the mechanism behind increased secretion despite reduced gland size in LGs, we examined mRNA expression of AQP5, a water channel essential for water secretion. There were no differences in the *Aqp5* mRNA expression levels between the control and *Cdc42*^cKO^ groups for either PGs or LGs (Figure 4a). However, the protein expression level of AQP5 was significantly decreased in *Cdc42*^cKO^ PGs, whereas it showed a tendency to increase in *Cdc42*^cKO^ LGs (Figure 4b), consistent with the water secretion data. Immunohistochemical analysis revealed that AQP5 expressed in acini was greatly reduced by *Cdc42*^cKO^ in PGs, but elevated in LG acini as a whole section trend (Figure 4c). Immunofluorescence staining was performed to assess more detailed AQP5 expression changes by observing the expression of AQP5 while checking the *Cdc42*^cKO^ acini. Comparing the fluorescence intensity of AQP5 in ducts and acini, AQP5 expressed in PG acini was reduced by *Cdc42*^cKO^. No changes were observed in PG ducts that did not express AQP5. In contrast, LG control slightly expressed AQP5 in acini compared to ducts which mainly expressed AQP5. The AQP5 in acini was enhanced by *Cdc42*^cKO^ (Figure 4d). These data suggested that *Cdc42*^cKO^ downregulated AQP5 protein in acinar cells in PGs but upregulated it in the LGs.

## 3. Discussion

In this study, we generated *Cdc42*^cKO^ mice to examine the roles of Cdc42 in the structure and physiological functions of mature salivary and lacrimal acinar cells. The results revealed that the acinar luminal structures of SGs and LGs were similarly expanded, and acinar cell apoptosis occurred. Intriguingly, water secretory functions were found to be oppositely affected, with a decrease in saliva production and an increase in tear production. Consistent with the results of water secretion, the AQP5 protein expression level in PGs of *Cdc42*^cKO^ mice was decreased compared to the control mice, while it was increased in the LGs. On the other hand, the *Aqp5* mRNA expression levels showed no significant differences. Furthermore, it appears that *Cdc42*^cKO^ alters the expression level of AQP5 in acinar cells in both PGs and LGs, although we cannot determine whether the effect is direct or indirect. Taken together, these findings indicate that *Cdc42*^cKO^ affects water secretion oppositely in SGs and LGs by modulating the AQP5 expression in salivary and lacrimal acinar cells in opposite directions (as illustrated schematically in Figure 5).

### 3.1. Differences in Water Secretion Function and AQP5 Expression between SGs and LGs

We found that the luminal structure of acinar cells was altered in both *Cdc42*^cKO^ SGs and LGs. Meanwhile, the saliva volume and AQP5 expression level were reduced in PGs, while the tear volume and AQP5 expression level were elevated in LGs. These results suggest that the volume of stimulated tear secretion was not dependent on the number of lacrimal acinar cells or the size of the LGs, at least in our case. AQP5 is widely known as a water channel, and it is used as an indicator of water secretion by both SGs and LGs in both humans and mice [16,17,18,19]. Consistent with previous reports [19,20], AQP5 in LGs was expressed in both acini and ducts, whereas in PGs it was expressed only in acini (Figure 4c). When comparing the acini and ducts within a single IHC image of an LG, most AQP5 in controls was expressed in ducts and AQP5 in acini was expressed slightly. *Cdc42*^cKO^ enhanced the intensity of AQP5 expression more in the acini than in the ducts. This suggests that the increased AQP5 expression in the acini may be responsible for the increased tear secretion. Therefore, we consider that the opposite phenomena of water secretion between SGs and LGs were due to their differences in AQP5 expression. In general, two pathways of water secretion are considered: transcellular and paracellular transport. Thus, we speculate that transcellular water transport in LGs was originally low due to low expression of AQP5, but *Cdc42*^cKO^ increased AQP5 expression of the acini.

### 3.2. Mechanism of Cdc42’s Regulation of AQP5 Expression

Post-translational modification, miRNA, and thyroid hormones were reported to regulate AQP5 protein expression [21]. Our data show a significant change in AQP5 protein expression, but not in *Aqp5* mRNA expression upon Cdc42 depletion (Figure 4a,b). It was reported that decreased expression of AQP5 due to ubiquitination may lead to decreased salivary secretion [22]. Based on these findings, we speculate that the post-translational modifications of AQP5 induced by *Cdc42*^cKO^—especially fluctuations in ubiquitination leading to proteolysis—are reflected in the changes in AQP5 protein expression. There is no report that Cdc42 directly regulates AQP5 expression, but active RhoA was reported to attenuate the effect of simvastatin-induced membrane accumulation of AQP2, another isoform of aquaporin [23]. It is likely that Rho family small GTPases are involved in the expression of aquaporin, though more extensive studies are needed to confirm this. Thus, the mechanism is still not fully understood, but there is speculation that *Cdc42*^cKO^ may have altered the acinar cell polarity, resulting in changes to the luminal structure, which, in turn, affects the membrane transport of AQP5, and activates AQP5 degradation systems (e.g., ubiquitination).

### 3.3. Acinar Cdc42 Deficiency Induced Apoptosis

The cell death induced by *Cdc42*^cKO^ in this study is likely apoptosis, which is not the same as necrosis. It is known that apoptosis does not induce inflammation and does not adversely affect surrounding cells [24,25]. Therefore, the effect of cell death in decreasing water secretion is considered to be minimal compared to the effect of necrosis. Cdc42 regulates various pathways, maintaining a balance between survival and death. While it is certain that abnormal Cdc42 expression is associated with apoptosis, whether Cdc42 acts in a pro- or inhibitory manner on apoptosis depends on the cell type and conditions. 

Mist1 expresses in serous and serous demilune acinar cells, but not in excretory ducts or mucous cells [26]. Consistent with the location of Mist1 expression, the number of apoptotic cells is elevated in acini but is unchanged in the ducts of SGs and LGs in *Cdc42*^cKO^ mice (Figure 2). This suggests that loss of Cdc42 in acinar cells acts to upset the balance between cell survival and death, resulting in apoptosis of acinar cells but not duct cells. Loss or dominant-negative of Cdc42 has been reported to be associated with increased apoptosis in several cell types, including podocytes [27], human T-cells [28], and bladder cancer cells [29]. Those reports support our experimental results.

Generally, apoptosis is regulated by pro-apoptotic factors such as Bax and anti-apoptotic factors such as Bcl-2, but the mechanism by which Cdc42 deficiency promotes exocrine acinar cell apoptosis requires further study. Taken together, in the SGs, both acinar cell apoptosis and decreased AQP5 expression are considered to be responsible for the decreased saliva secretion. On the other hand, in the LGs, acinar cell apoptosis is induced, but increased AQP5 expression in the remaining acinar cells may have counteracted this negative effect and led to increased tear secretion. Although not examined in this study, other possible factors include changes in transporter expression and function, changes in muscarinic receptor expression, bloodstream, and changes in AQP5 expression in the ducts.

### 3.4. Cdc42^cKO^ Mice as a Potential Mouse Model for Aging Exocrine Glands

The *Cdc42*^cKO^ mice used in this study were characterized by lowered saliva and elevated tear secretions (Figure 3). It was reported that saliva decreased while tears increased in aging mice [30,31], and a tendency for extension of the lumen was observed in the elderly [32]. Moreover, dysregulation of Cdc42 was reported to be associated with aging [33]. Several studies reported decreased expression and activity of Cdc42 with aging [34]. Therefore, *Cdc42*^cKO^ mice have potential for use as a mouse model of exocrine gland senescence.

## 4. Materials and Methods

### 4.1. Materials

The following materials were used in this study: KAPA Express Extract Kit (KK7102, Roche Diagnostics, Mannheim, Germany), EmeraldAmp Max PCR Master Mix (RR320A, Takara Bio, Shiga, Japan), tamoxifen (T5648, Sigma-Aldrich, St. Louis, MO, USA), corn oil (C8267, Sigma-Aldrich, MO, USA), medetomidine (Dorbene^®^, Kyoritsu Seiyaku, Tokyo, Japan), midazolam (SANDOZ, Tokyo, Japan), butorphanol (Vetorphale^®^, Meiji Seika Pharma, Tokyo, Japan), RNAlater^®^ Stabilization Solution (AM7020, Thermo Fisher Scientific, Waltham, MA, USA), FavorPrep Tissue Total RNA Mini Kit (FATRK 001, Chiyoda Science, Tokyo, Japan), ReverTra Ace qPCR RT Master Mix (FSQ-201, TOYOBO, Osaka, Japan), SYBR Premix Ex Taq II (RR820A, Takara Bio, Shiga, Japan), primer pairs (Eurofins Genomics, Tokyo, Japan), phalloidin-647 (ab176759, Abcam, Cambridge, UK), RIPA lysis buffer (WSE-7420, ATTO, Tokyo, Japan), protease inhibitor cocktail (05892791001, Roche Diagnostics, Mannheim, Germany), phosphatase inhibitor cocktail (04906837001, Roche Diagnostics, Mannheim, German), Bio-rad Protein Assay Dye Reagent (Cat#500-0006, Bio-Rad Laboratories, Hercules, CA, USA), bovine serum albumin (Cat#A7906, Sigma-Aldrich, MO), Can Get Signal^®^ Immunoreaction Enhancer Solution (F0991K, TOYOBO, Osaka, Japan), stripping buffer (BW-6010, Apro Science, Tokushima, Japan), DAB Substrate kit (SK-4100, Vector Laboratories, Newark, CA, USA), and DAPI (NM167, Dojindo Laboratories, Kumamoto, Japan).

### 4.2. Mouse Strains

*Cdc42^f^*^l/fl^/mTmG^+/+^ mice were kindly provided by Y. Zheng (Cincinnati Children’s Hospital Medical Center, Cincinnati, OH, USA). *Cdc42*^fl/fl^/mTmG^+/+^ mice contain a loxP site flanking the second exon of *Cdc42*, and the mice were ubiquitously labeled using a Rosa26mTmG reporter. mTmG is a cell membrane-targeted, two-color fluorescent Cre-reporter. Before Cre recombination, cell membrane-localized tdTomato (mT) fluorescence expression is widespread in cells/tissues. Cre recombinase-expressing cells have cell membrane-localized EGFP (mG) fluorescence expression replacing the red fluorescence [35]. *Cdc42*^fl/fl^/mTmG^+/+^ mice were first crossed with *Mist1*^CreERT2/CreERT2^ mice (RRID: IMSR_JAX:029228) purchased from The Jackson Laboratory (Bar Harbor, ME, USA). *Mist-1*^CreERT2^ mice express CreERT2 exclusively in acinar cells. *Mist1*^CreERT2/CreERT2^/*Cdc42*^fl/fl^/mTmG^+/+^ mice were produced by repeated crossbreeding. *Mist1*^CreERT2/CreERT2^/*Cdc42*^fl/fl^/mTmG^+/+^ mice and *Cdc42*^fl/fl^/mTmG^+/+^ mice were crossed to create *Mist1*^CreERT2/+^/*Cdc42*^fl/fl^/mTmG^+/+^ mice with *Mist1* heterozygosity for experiments (Figure 1a). Mice were euthanized by cervical dislocation following exposure to carbon dioxide. The genetic background of these mice was C57BL/6. Animals were bred and mated at Asahi University (Gifu, Japan) according to approved standards, with controlled temperature (22 °C) and lighting (14 h light, 10 h dark), plus unlimited food and water. Temperature and humidity were controlled at 22–26 °C and 50–65%, respectively.

### 4.3. Tamoxifen Injection

Tamoxifen was dissolved in corn oil for 1 h at 65 °C to prepare 20 mg/mL tamoxifen in corn oil. The 8-week-old *Mist1*^CreERT2/+^/*Cdc42*^fl/fl^/mTmG^+/+^ mice were anesthetized with isoflurane and treated with tamoxifen (226 mg/kg) or the same volume of corn oil by subcutaneous injection (s.c.) once every 2 days for a total of 2 doses and used for experiments 4 weeks after administration.

### 4.4. Anesthesia Administration

Mice were anesthetized via an intraperitoneal (i.p.) injection of an anesthetic agent mixture (0.75 mg kg^−1^ medetomidine, 4.0 mg kg^−1^ midazolam and 5.0 mg kg^−1^ butorphanol) at a volume of 0.05 mL per 10 g body weight [36].

### 4.5. RNA Extraction and Real-Time RT PCR (qPCR) Analysis

Mice were euthanized with carbon dioxide, and their PGs and LGs were isolated. The isolated glands were soaked in RNAlater^®^, which prevents their degradation by RNase. Total RNA was purified from each whole gland using a Tissue Total RNA Purification Mini Kit after eliminating DNA contamination with DNase I, according to the manufacturer’s instructions. cDNA was synthesized from 0.5 μg total RNA by reverse transcription with oligo dT primers using ReverTra Ace qPCR RT Master Mix. The template cDNA was amplified using SYBR Premix Ex Taq II and specific primer pairs in a Thermal Cycler under the following conditions: 40 cycles of 95 °C for 5 s and 60 °C for 30 s after initial denaturing at 95 °C for 30 s. Melting curve data were obtained by increasing the temperature from 60 to 95 °C. Gene expressions were quantified using a standard curve, and then genes were normalized to *Gapdh* as an internal control. Reactions were run in duplicate. The primer pairs used are shown in Table 1. Three mice were used for the control group and four mice were used for the *Cdc42*^cKO^ group.

### 4.6. Immunofluorescent Staining and Imaging

Mice were anesthetized by the methods described above, and transcardiac perfusion fixation with 4% PFA in PBS was performed. Glands were extracted and immersed overnight in 4% PFA in PBS. For preparation of frozen mouse tissue sections, the tissues were placed in 15% sucrose in PBS overnight and then 30% sucrose in PBS overnight, and then embedded with OCT compound to freeze. Cryosections (10 μm thick) were prepared with a Leica CM3050 and stained for histological analysis. Thin sections were incubated in blocking solution (10% fetal bovine serum, 0.02% saponin and 0.02% sodium azide in PBS) containing phalloidin-647 and DAPI for 30–60 min at room temperature. When stained with AQP5, the sections were incubated overnight at 4 °C in a blocking agent with AQP5 antibodies and were then washed with PBS, following incubation with anti-rabbit IgG-647 for 30 min. The sections were washed and incubated with DAPI for 30 min. The sections were washed 3 times with PBS for 5 min. Finally, samples were mounted on a glass slide and covered with a coverslip. Fluorescence from the staining or mT/mG was acquired using LSM710, a laser confocal microscope (Carl Zeiss, Tokyo, Japan). All images were compiled using Fiji/ImageJ (ver. 1.52p) (Rockville, MD, USA).

### 4.7. Measurement of Gland Weight

Mice were euthanized with carbon dioxide. Their PGs, SMGs, SLGs, and LGs were isolated, photos were taken, sizes were compared, then gland weight was measured.

### 4.8. In Situ Apoptosis Analysis

Terminal deoxynucleotidyl transferase dUTP-biotin nick end labeling (TUNEL) assay was performed on frozen sections. Apoptotic DNA double-strand breakages were labeled with streptavidin-Alexa Fluor^®^ 647 after treatment with terminal transferase and biotin-16-dUTP. Fluorescence images were captured using a Carl Zeiss LSM710, and TUNEL-positive cells were counted as follows. Three images were taken with a 20× objective of different fields of view, and the average number of positive cells in those fields was used as the value for one section. Three mice were used for each group, and the calculated mean values were standardized to the control.

### 4.9. Measurement of Pilocarpine-Induced Saliva and Tear Volumes

Mice were anesthetized (hypothermia during anesthesia was prevented by using heating pads) by the methods described above, and pilocarpine-induced saliva and tear volumes were determined, as we described previously [36]. Briefly, the secreted saliva was absorbed into paper plugs inserted into the oral cavity after pilocarpine was injected i.p. at 0.5 mg/kg. Simultaneously, the tear volume was measured in the same mice by carefully placing a phenol red-impregnated thread at the canthus. Both saliva and tears were collected for 30 s every 2 min until 20 min after pilocarpine injection. The total saliva weight (mg) and tear volume (mm) were calculated by adding up each value. Three mice were used for the control group and four mice were used for the *Cdc42*^cKO^ group.

### 4.10. Western Blot Analysis

Mice were euthanized with carbon dioxide, and their PGs and LGs were isolated. The glands were homogenized in ice-cold RIPA lysis buffer containing a protease inhibitor cocktail and a phosphatase inhibitor cocktail. The homogenates were incubated on ice for 15 min, and then cleared lysates were obtained by centrifugation (14,000× *g* for 10 min at 4 °C). The protein concentration in each lysate was determined by the method of Bradford using the Bio-rad Protein Assay Dye Reagent and bovine serum albumin as a standard. Equal amounts of protein (10–20 μg) were separated by SDS-PAGE under reducing conditions with a Mini-Protean 3 Cell system (Bio-Rad Laboratories, CA). After electrophoresis, the separated proteins were transferred onto PVDF membranes using a Trans-Blot Turbo System (Bio-Rad Laboratories, CA). Membranes were blocked with 10% skim milk in 10 mM Tris-buffered saline (pH 7.4) containing 0.1% Tween20 (TBS-T) for 1 h at room temperature and then probed with a primary antibody using a rapid immunoblot method, as previously described [37,38]. Briefly, the primary antibody was diluted in 2.8 mL of 10% Can Get Signal^®^ Solution 1/90% water and 0.2 mL of 5% skim milk in TBS-T, and secondary antibodies were diluted in 2.3 mL of 10% Can Get Signal^®^ Solution 2/90% water and 0.7 mL of 5% skim milk in PBS-T. The antibodies used in this study and details are shown in Table 2. Signals were visualized using Amersham ECL Western Blotting Detection Reagent (Cat# RPN2235, Cytiva, Tokyo, Japan) and Light-Capture II (ATTO, Tokyo, Japan). The signal intensities of the captured images were analyzed with CS Analyzer 3.0 (ATTO, Tokyo, Japan). When deemed necessary, the blots were re-probed with different antibodies after stripping with stripping buffer in accordance with the manufacturer’s protocol. Three mice were used for the control group and four mice were used for the *Cdc42*^cKO^ group.

### 4.11. Immunohistochemistry

After the above fixation, frozen tissues were cut into 10 μm thick sections. Sections of the glands were stained with AQP5 antibody and visualized using the DAB Substrate kit. Images were acquired with an optical microscope (BX51, Olympus, Tokyo, Japan).

### 4.12. Statistical Analysis

Data are presented as the mean ± standard deviation (SD). Statistical comparisons were made using a two-tailed Student’s *t* test (Figure 1b,c, Figure 2a,b, Figure 3b,d and Figure 4a,b) and two-way ANOVA followed by Sidak’s multiple comparisons test (Figure 3a,c). Values (P) below 0.05 were regarded as statistically significant differences. These statistical analyses were performed using GraphPad Prism7 (GraphPad Software, La Jolla, CA, USA).

## 5. Conclusions

This study is the first to demonstrate that loss of Cdc42 in exocrine acini decreases saliva secretion but increases tear secretion, even though Cdc42 similarly maintains acinar cells in both SGs and LGs. Our findings are an important step forward in understanding the physiological functions of Cdc42 in live mice, and by extrapolation, in humans. Decreased saliva due to dysfunction of SGs can lead to periodontal disease and dental caries in humans, but no curative therapy has yet been established. Identifying factors that regulate increased tear secretion in LGs of *Cdc42*^cKO^ mice may provide clues to developing a fundamental treatment for SG dysfunction in humans.

## Figures and Tables

**Figure 1 ijms-24-17220-f001:**
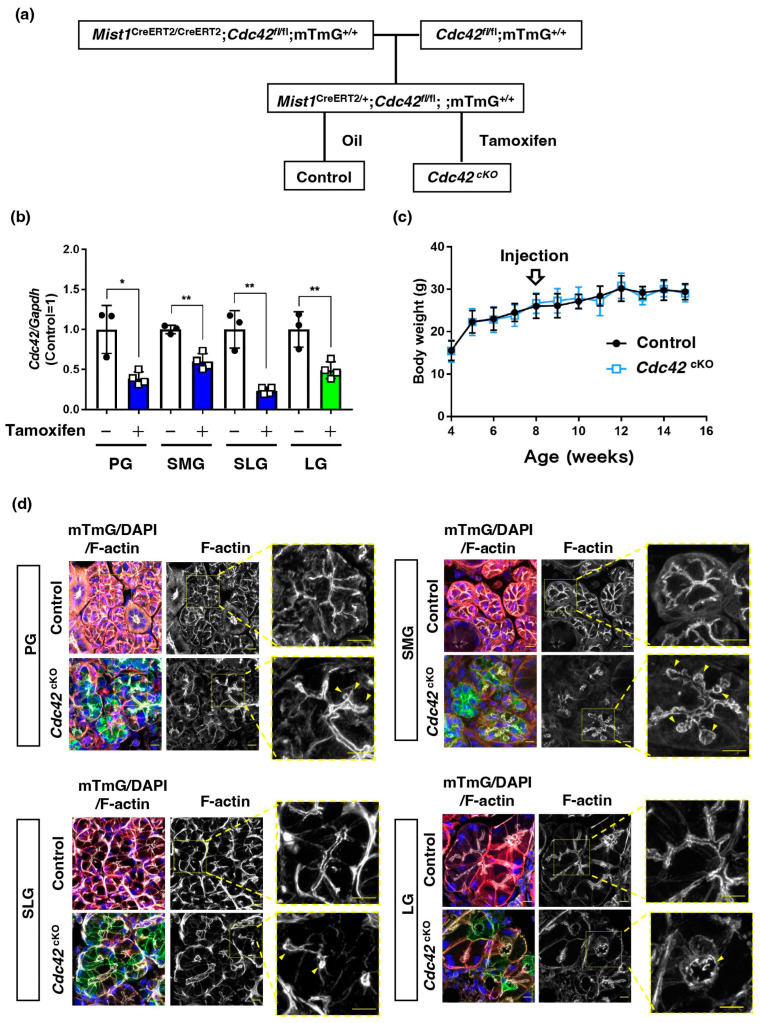
Acinar cell-specific *Cdc42* conditional knockout in mice altered the luminal structure in SGs and LGs. (**a**) Schematic of the generation of experimental mice. *Mist1*^CreERT2/+^/*Cdc42*^fl/fl^/mTmG^+/+^ mice can be used as exocrine acinar cell-specific conditional KO (*Cdc42*^cKO^) mice induced by tamoxifen administration. (**b**) Quantitative RT-PCR analysis shows reduced expression of *Cdc42* genes in the PGs, SMGs, SLGs, and LGs of tamoxifen-treated mice. Blue bars indicate *Cdc42*^cKO^ SGs and green bar indicates *Cdc42*^cKO^ LGs. Results are shown as mean ± S.D.; * *p* < 0.05; ** *p* < 0.01. Three mice were used for the control group and four mice were used for the *Cdc42*^cKO^ group. (**c**) Body weight of male mice before and after corn oil or tamoxifen administration. (**d**) Frozen sections of PGs, SMGs, SLGs, and LGs of control or tamoxifen-treated mice were stained with phalloidin-647 (F-actin, lumen) and DAPI (nucleus) viewed with confocal microscopy. Membrane-tagged tdTomato (mT: red) was detected ubiquitously in the control mice. Tamoxifen treatment induced expression of membrane-tagged GFP (mG: green) in acinar cells. mG-labeled acini in *Cdc42*^cKO^ mice showed expanded lumen and swelling of those tips. Yellow arrowheads indicate an altered luminal structure. Scale bars are 10 µm.

**Figure 2 ijms-24-17220-f002:**
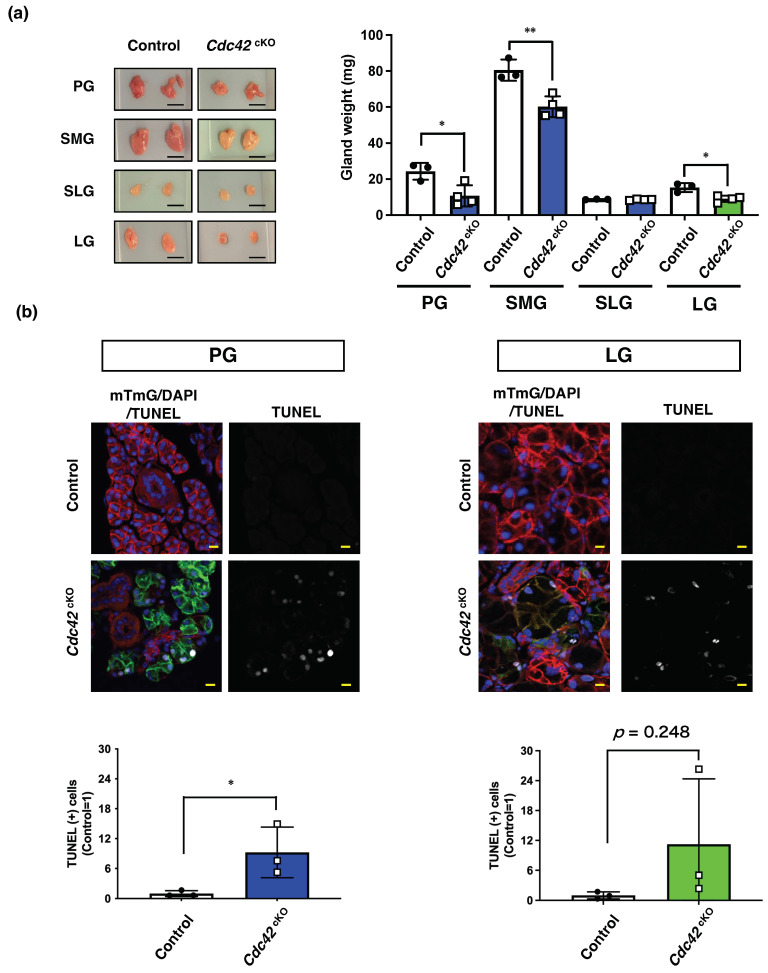
*Cdc42*^cKO^ reduces the number of acinar cells in exocrine glands by inducing apoptotic cell death. (**a**) Representative images of exocrine glands isolated from control and *Cdc42*^cKO^ mice. Scale bars indicate 5 mm. The tissue weights of glands from 12-week-old mice were measured. Results are shown as mean ± S.D.; * *p* < 0.05; ** *p* < 0.01 Three mice were used for the control group and four mice were used for the *Cdc42*^cKO^ group. (**b**) *Cdc42* ^cKO^ induced acinar cell apoptosis. Apoptotic cells of PGs and LGs in frozen sections were detected by TUNEL staining. Scale bars are 10 µm. Bar graphs indicate the relative number of TUNEL (+) cells. Results are shown as mean ± S.D.; * *p* < 0.05.

**Figure 3 ijms-24-17220-f003:**
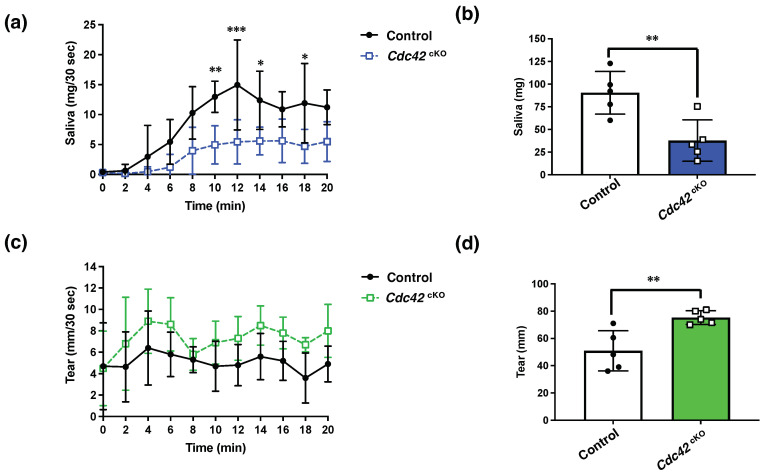
Cdc42 depletion decreased pilocarpine-stimulated saliva secretion, but increased tear secretion. Saliva and tear volumes were measured. (**a**) Saliva volume every 30 s (mg/30 s). (**b**) Total saliva volume (mg). (**c**) Tear secretion volume every 30 s (mm/30 s). (**d**) Total tear volume (mm). Results are shown as mean ± S.D.; * *p* < 0.05; ** *p* < 0.01; *** *p* < 0.001. Each group used five mice.

**Figure 4 ijms-24-17220-f004:**
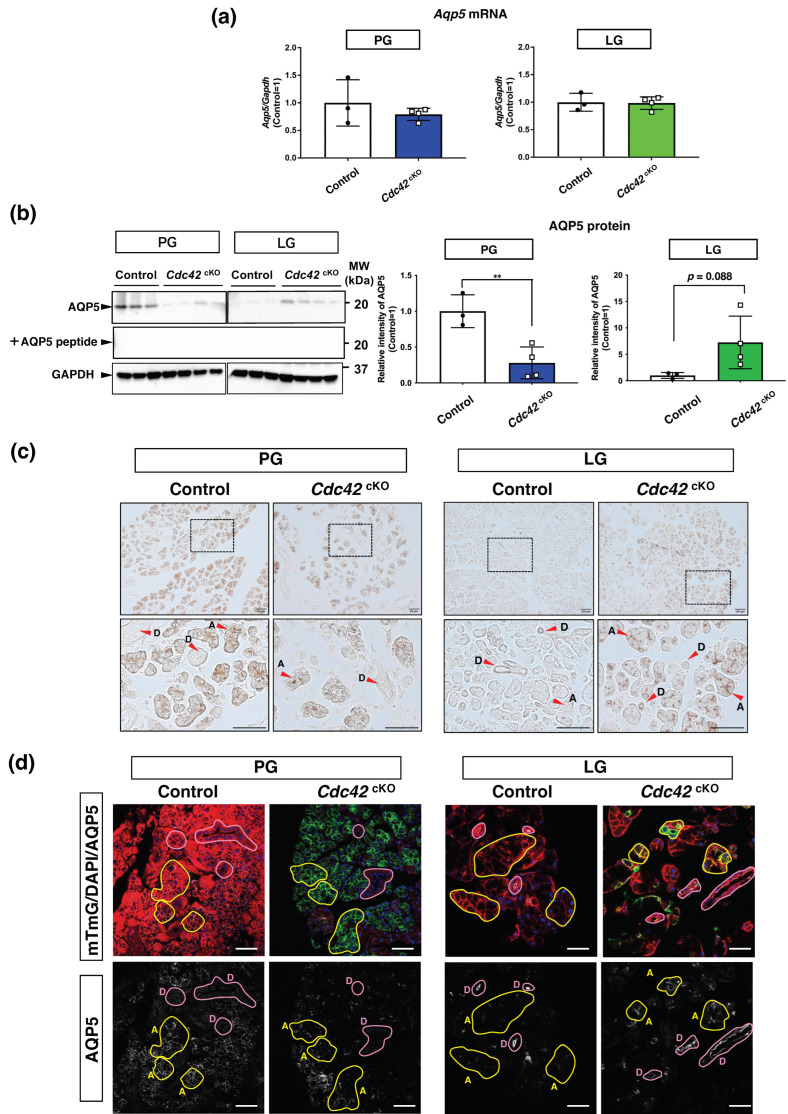
AQP5 protein in acinar cells is downregulated by *Cdc42*^cKO^ in PGs but upregulated in LGs. AQP5 expression analysis. (**a**) Quantitative RT-PCR analysis shows *Aqp5* mRNA expression was not significantly different between the control and *Cdc42*^cKO^ groups. Bar graphs indicate relative expression of *Aqp5/Gapdh*. Three mice were used for the control group and four mice were used for the *Cdc42^c^*^KO^ group. (**b**) Western blotting with anti-AQP5 and anti-GAPDH. For negative control of AQP5 immunoblotting, AQP5 antibody was preincubated with five times the amount of AQP5 peptide for 1 h at room temperature and used as the primary antibody of Western blotting (+AQP5 peptide). Bar graphs indicate relative intensity of AQP5. Three mice were used for the control group and four mice were used for the *Cdc42^c^*^KO^ group. (**c**) Immunohistochemical analyses of AQP5 protein. A: acini, D: ducts. Scale bars: 100 µm. (**d**) Immunofluorescent staining of AQP5 protein. A: acini, D: ducts. Scale bars: 50 µm. Results are shown as mean ± S.D.; ** *p* < 0.01.

**Figure 5 ijms-24-17220-f005:**
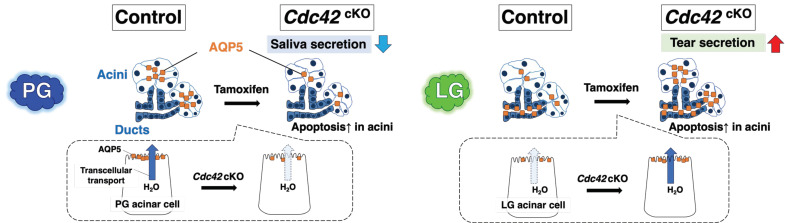
Overview of this study. *Cdc42*^cKO^ altered the acinar cell polarity, which caused changes in the luminal structure, leading to defects in membrane transport of AQP5 and promotion of AQP5 degradation systems (e.g., ubiquitination). In PGs, saliva volume was reduced by both apoptosis and AQP5 down-regulation. In LGs, tear volume was increased by AQP5 up-regulation on the apical membrane of acini.

**Table 1 ijms-24-17220-t001:** Primer sequences for expression studies.

Gene Name	Direction	Primer Sequence (5′−3′)	Annealing Temperature (°C)	Reference
*Cdc42*	Forward	GTG TGT TGT TGT TGG TGA TGG T	60	This study
	Reverse	AGT CCA AGA GTG TAT GGC TCT		
*Aqp5*	Forward	CAT CTT CTA CGT GGC AGC CC	60	This study
	Reverse	ATT AAC TCC ACC ACC ACG GC		
*Gapdh*	Forward	AGG CCG GTG CTG AGT ATG TC	60	
	Reverse	TGC CTG CTT CAC CAC CTT CT		

**Table 2 ijms-24-17220-t002:** Antibodies for expression studies.

Antibody	Species	Dilution for WB	Dilution for IF, IHC	Campany (Catalog#)
AQP5	pAb, Rb	1:6K	1:200, 1:300	Millipore (AB15858)
GAPDH	pAb, Rb	1:20K	−	Gene Tex (GTX100118)
Rb IgG-F(ab’)2-HRP	pAb, Gt	1:20K	1:500	Millipore (AQ132P)
Rb IgG H&L-647	pAb, Donkey	−	1:300	abcam (ab150075)
Streptavidin-HRP	−	−	1:1K	abcam (ab7403)

## Data Availability

Data is contained within the article.

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
