# Peer review of "Loss of Cdc42 in Exocrine Acini Decreases Saliva Secretion but Increases Tear Secretion—A Potential Model of Exocrine Gland Senescence"

_ijms, 2023, doi:10.3390/ijms242417220_

Round 1

Reviewer 1 Report

Comments and Suggestions for Authors

Dear Author, 

Please find my Comments attached. 

Regards

Comments on the Quality of English Language

Satisfactory

Reviewer 2 Report

Comments and Suggestions for Authors

The manuscript investigates the effects of a conditional knock-out of Cdc42 on salivary and lacrimal gland size and morphology.  Parotid and lacrimal gland secretory function and Aqp5 expression are compared in Cdc42 k/o vs control mice.  PG, SMG and LG weight were decreased in Cdc42 k/o mice associated with increased   apoptosis in the PG, but apoptosis was variable in LG’s from k/o mice.  PG secretion decreased but LG secretion increased in k/o mice.  Changes in Aq5 protein expression (assessed by w/blotting) followed the same direction as altered secretion but was not statistically significant in the LG.  IHC distribution of Aq5 was altered the examples presented.

Comments.

1.     The IHC shown in figure 4c is puzzling.  Aqp5 positivity is evident in the control PG and the Cdc42 k/o LG and a reduced intensity in the Cdc42 k/o PG, however there is little evidence of Aqp5 expression in the control LG.    The authors should repeat this staining to increase the positive signal and show more than one example.  Visualization by fluorescence would be more sensitive. 

2.     LG ductal expression should be indicated in fig. 4c.

3.     In relation to the interpretation the discussion states:  On the other hand, in the LGs, acinar cell apoptosis is induced, but increased AQP5 expression in the remaining acinar cells may have counteracted this negative effect and led to increased tear secretion.  How might Aqp5 expression in LG ductal cells influence secretion?  The authors should comment on this.

4.     Secretion of saliva and tears is shown in Fig.3.  What was the secretion per unit gland size?  This would provide a more meaningful comparison and may modify the interpretation of results.

5.     In 3.3 of the Discussion, the potential role of inflammation is considered.  Was there any evidence of inflammatory cell infiltration?  What methods were used to assess potential inflammation?

6.     Frequent mention is made of expanded luminal structures, but the authors should indicate explicitly how they interpret this observation.

Round 2

Reviewer 2 Report

Comments and Suggestions for Authors

The authors have satisfactorily answered my concerns.